# Bridge Inspection with an Off-the-Shelf 360° Camera Drone

Andreas Humpe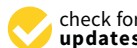

Campus Schachenmeierstrasse 35, University of Applied Sciences Munich, 80636 Munich, Germany;
humpe@hm.edu; Tel.: +49-89-1265-2112

**Abstract:** The author proposes a new approach for bridge crack detection by a 360° camera on top of a drone. Traditionally, bridge inspection is performed manually and although the use of drones has been implemented before, researchers used standard high definition cameras underneath the drone. To make the approach comparable to the conventional approach, two bridges were selected in Germany and inspected for cracks and defects by applying both methods. The author follows an engineering design process and after developing a prototype of the drone with a 360° camera above the body of the drone, the system is built, tested, and the bridges are inspected. First, the critical parts of the bridges are inspected with an off-the-shelf drone with a high definition camera underneath the drone. The results provide a benchmark for comparison. Next, the new approach to bridge inspection by using a 360° camera on top of the drone is tested. The images of the critical parts of the bridge that were taken with the 360° camera on top of the drone are analyzed and compared to the images of the conventional approach with the camera underneath the drone. The results show that a 360° camera can be used for crack and defect detection with similar results to a standard high definition camera. Furthermore, the 360° camera is more suitable for inspecting corners or the ceiling of, e.g., an arch bridge.

**Keywords:** bridge inspection; 360° camera; defect detection; off-the-shelf drone

---

## 1. Introduction

According to Gkoumas et al. [1], there are more than 1234 km of road bridges with a length of over 100 m in Europe. The US Bureau of Transportation Statistics reports 614,386 bridges in the US at the end of 2016 [2]. Overall, there is a large number of bridges in use today. As such, bridges build an integral part of the road infrastructure worldwide and are seen as the most vulnerable element of the road network [3].

Most bridges in Western Europe were built in the aftermath of World War II during the 1950s, 1960s, and 1970s, with a peak in construction activity during the 1970s (for a discussion see [4,5]). As a result, many bridges are 40 to 50 years old.

The deterioration of a bridge depends on time, the nature of the environment, load capacity, construction procedures, load frequency, and quantum [6]. Many bridges that were built during the 1950s or 1960s in Germany suffer from corrosion and cracks. The summer temperature in Germany might be above 28 degrees; whereas winter temperatures can fall below minus 10 degrees. During winter; streets are usually treated with salt to avoid ice on the surface. This, in combination with the large temperature swings, puts severe stress on the surface and inner parts of a bridge and might cause defects over the years. The increase in extreme weather conditions in recent years, has put even more temporary stress on critical infrastructure, as documented by Groenemeijer et al. [7]. For instance, Woodward et al. [3] estimate that 37% of German and 30% of UK bridges have defects like corrosion of reinforcement, design/construction faults, faulty bearings, joints, or drainage.

Defects in bridges might have severe consequences ranging from traffic restrictions to collapse. Most recently, on the 14 August 2018, the viaduct over the Polcevera river in Genoa collapsed, killing 43 people [8]. The bridge was designed by Riccardo Morandi and opened to the public in 1967. The increasing problem of deteriorating bridges and the lack of sufficient maintenance is well documented in the literature (see inter alia [6,9–11]). Maintenance of old bridges is complicated and expensive. The traditional method of bridge inspection relies on workers going to the bridge and employing ladders, scaffolding, or lifters to reach the critical parts of the bridge that are not easily accessible [12]. This is very expensive and time-consuming [11]. According to [3], the direct cost of national bridge maintenance and repair in the UK is 180 million euros, in France, 50 million euros and in Norway, 30 million euros. However, national bridges only represent 10% of bridges in the UK and France and about 50% in Norway. Given the large number of bridges built during the 1970s, these costs are likely to increase even further when those bridges begin to deteriorate. In the US, the average inspection cost per bridge ranges from 4500 to 10,000 USD and might require closing lanes for one to three days for inspection [13]. Bridges are complicated structures with many poorly accessible areas [10]. This makes inspection even more complicated. As a result, manual bridge inspection is labor-intensive and risky. Generally, the inspection requires the bridge to be closed for traffic, inducing additional costs and inconvenience for travelers.

Hence, drone pre-inspection might lower costs and increase the efficiency of the inspection process. Ideally, a drone would take high-resolution pictures from critical parts that can be analyzed in the office. Only suspicious parts would then be inspected locally, whereas other parts are fine and need no further checks. In this context, it should be noted that only some tasks in bridge inspection are based on visual observation of the bridge. This includes the detection of cracks, rebar corrosion, or delamination. Nevertheless, there are other tasks that require direct contact of a sensor or device with the bridge surface [12]. Nonetheless, many bridge inspection tasks can be performed with a drone in combination with a high definition camera. There are many requirements for an autonomous drone inspection system. Besides the necessity to take high-resolution pictures from different angles, the coverage range of a drone and the stability of the system in bad environmental conditions (e.g., rain or wind) are key factors [10]. Furthermore, optimal inspection points and the distance between the camera and bridge parts must be established [14].

The use of drones for bridge inspection might be cost-effective, cause minimal disturbance, and reduce safety-related issues for manual inspectors. As a result, many studies have investigated the use of drones for bridge inspection. However, critical parts of a bridge include corners, the ceiling, or round arches. For those areas, it is difficult to take a picture with a camera that is positioned underneath a drone. One solution might be to make a drone fly at different angles than its usual horizontal flight in a steady manner, making sure that imaging can be performed. Instead, the author proposes the use of a 360° camera positioned above the drone. A camera underneath the drone is very well suited to take images of surfaces in front or below the drone, but makes it hard or even impossible to access a ceiling. In contrast, a 360° camera on top of a drone can take pictures in all directions, including the ceiling [15]. The 360° image can also be used in a VR setting at the office desk and enables convenient inspection of all recorded areas by "looking around". The use of a 360° image also simplifies the optimal flight and inspection strategy due to the 360° view. Given the shape of the bridge, the critical areas must still be identified, but the 360° image automatically provides front pictures of sloping, round and straight parts of the bridge.

This study is limited to two small bridges that are hardly used anymore in Germany. Hence, results cannot be generalized for larger bridges that are in heavy use, where it might be much more difficult to access critical parts and protect people and objects during the inspection. However, for safety reasons, the author decided only to investigate small discarded bridges. As many bridges are inspected by local authorities with budget constraints, the focus is on a cheap off-the-shelf solution for the major parts (e.g., the drone and cameras). More expensive equipment and special-purpose vehicles for bridge inspection have been investigated before, and the study adds to the field by applying a cheap and easy to assemble drone

inspection system with an additional 360° camera on top of the vehicle. One of the inspected bridges is an arch bridge, of which relatively limited research has been made. The article can show that the inspection system is able to detect defects and cracks in the selected bridges. The main objective of the paper is to define the position and the kind of camera to use in the prototype for bridge inspection. Furthermore, the article investigates whether a 360° camera on top of a drone can produce images for automatic crack detection comparable to images from a standard HD camera underneath the drone.

## 2. Materials and Methods

To build a prototype of a drone that is equipped with a 360° camera, an engineering design process is applied. To make use of an off-the-shelf drone in combination with a 360° camera for bridge inspection, an appropriate drone and camera must be selected. The camera must be attached to the drone in a simplistic and reversible way. The specific requirements of the equipment for visual bridge inspection must be specified, a possible solution chosen, and a prototype developed. Afterward, the prototype is tested, and if the solution does not meet all requirements, the prototype must be adjusted and re-tested (for a discussion of the engineering design process, see [16]).

To visually detect cracks and defects in bridges, images were manually and automatically inspected. The author hypothesized that both the standard high definition (HD) camera and the 360° degree camera were generally able to take images for crack detection with comparable results. Furthermore, the author hypothesized that a 360° camera on top of the drone was superior to a camera underneath the drone for inspecting arches, corners, and the ceiling of a bridge. To test the hypotheses, images of different parts of the bridge were taken by both cameras, and results were analyzed and compared.

### 2.1. Case Study I

For the investigation, it was necessary to select an old bridge with various defects but little traffic for safety reasons. The first bridge that satisfied the criteria was an old bridge located in a town called Thambach in Lower Bavaria, Germany. The bridge had a concrete deck for car traffic and was built on a round arch structure. There were rail tracks underneath the bridge, but nowadays, the tracks were hardly used. Unfortunately, it was not possible to ascertain the date of construction. However, the bridge was in poor condition, and various defects were visible. Figure 1 shows the front of the selected bridge in Thambach:

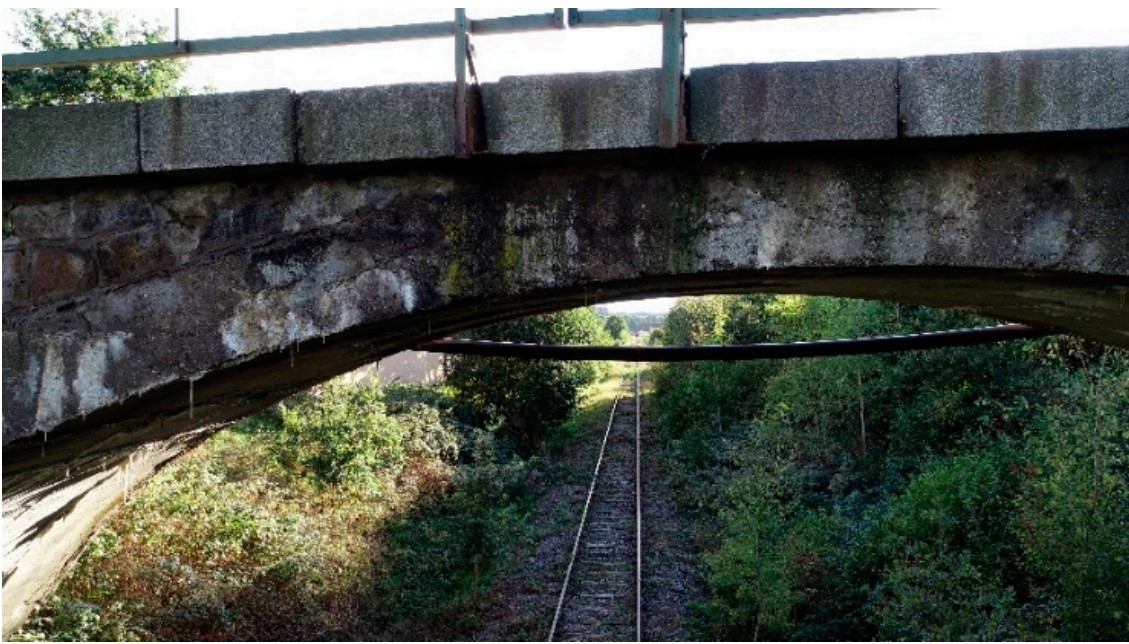

**Figure 1.** Front view of the selected bridge in Thambach, Germany.

## 2.2. Case Study II

The second bridge that satisfies the criteria was also an old bridge located in a town called Gangkofen in Lower Bavaria, Germany. The bridge had a steel deck with railroad tracks for train traffic and was built on 2 columns. There were car lanes underneath the bridge with little traffic and a river next to the street. It was not clear when the bridge was built, but it was also in poor condition with various visible defects. Figure 2 shows the front of the selected bridge in Gangkofen:

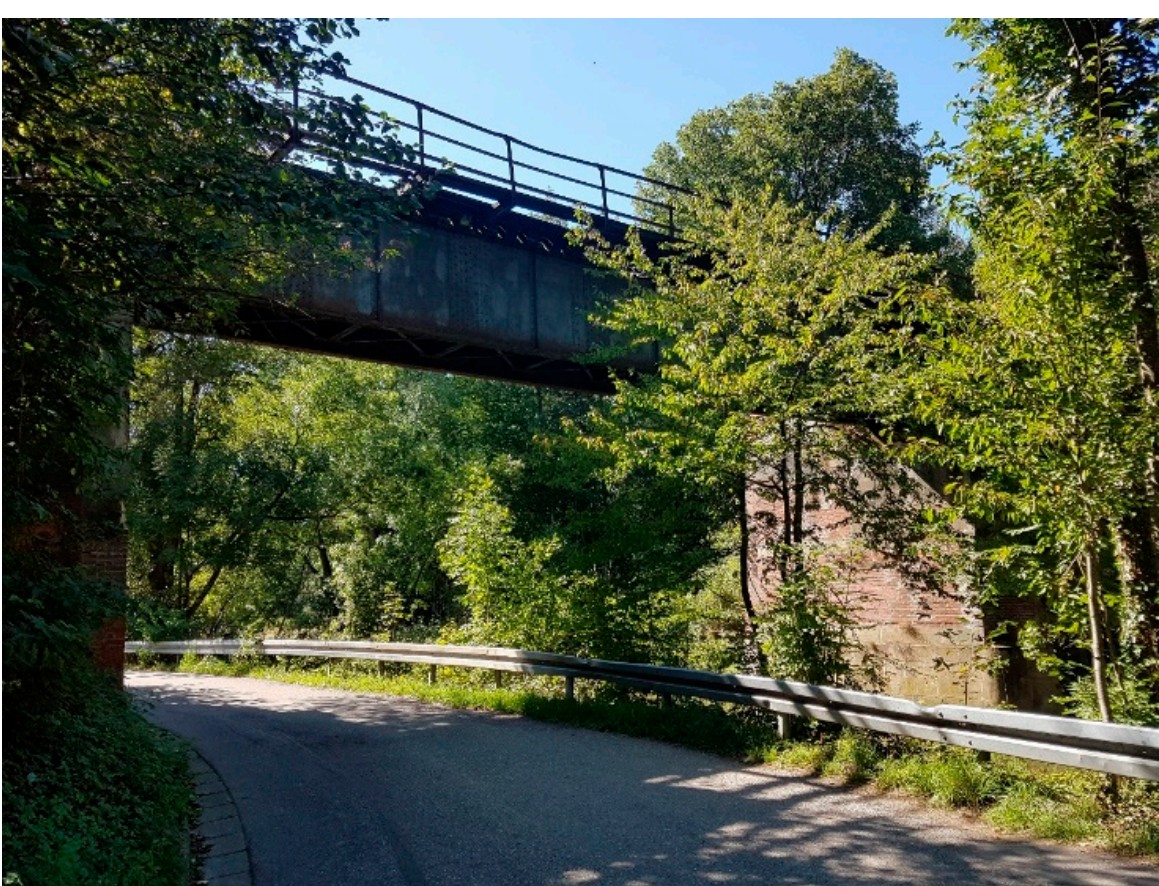

**Figure 2.** Front view of the selected bridge in Gangkofen, Germany.

## 2.3. Data Collection

For data collection, the author went to the bridges with the modified drone for crack inspection. A bridge inspection methodology was developed by [10] and followed 5 stages. It was recommended to start with a bridge information review. This included the analysis of building plans or historical inspection reports. In the next step, a thorough site risk assessment of the bridge was necessary. For instance, nearby trees or traffic lanes might induce high risk for drone-enabled inspections. The 3rd step consisted of the drone pre-flight setup. A thorough inspection of the drone and software was recommended. In step 4, the actual bridge inspection by the drone was performed. Finally, the collected information was used for damage identification of critical parts of the bridge.

A literature and web search helped to identify different drone platforms that can generally be used for bridge inspection. It was critical to use a platform that was durable and could be used for industrial applications. The appropriate drone must exhibit a stable flight behavior and must have a decent flight time for operations. Similarly, the maximum operating distance was of high relevance. Hence, the flight weight, efficiency of the vehicle, and battery duration were critical. Furthermore, the platform must allow for autonomous high precision navigation for any autonomous flight and the use of different sensors (in particular, a high-resolution camera that can be rotated).

## 3. Prototype Design

There were a series of requirements and constraints for a drone to perform bridge inspection tasks. As bridges might be large and there are many critical parts of a bridge that require inspection, it is important that the drone has a reasonable flying time. Before the actual inspection starts, there is some additional battery capacity needed for safety checks and flying time to critical parts. However, there is a trade-off between flying time, size, and cost of a drone. Floreano et al. [17] report a positive relationship between mass and flight time of drones.

Although larger drones tend to have longer flight time, they have several disadvantages as well. Larger drones are also more regulated than small ones, usually much more expensive, and might not be able to fly close enough to critical parts of a bridge. Therefore, there is a trade-off between flight time, size, and cost. Therefore, the author set an appropriate flight time of the drone between 25 and 30 min. Due to the author's budget constraint, the price limit was set at 2500 USD.

In order to take images of defects for manual and automatic crack detection, the on-board camera underneath the drone must be of reasonable quality. As an off-the-shelf drone was used, in order to obtain a reasonable resolution, a 4K camera was necessary for bridge inspection. Furthermore, the camera should be able to take images of parts below the bridge deck with little light. The camera, therefore, needs decent resolution with low illumination.

As the drone must carry the additional 360° camera, sufficient payload capacity was necessary as well. A reasonable payload capacity to carry any additional equipment was set between 400 and 500 g. Again, bearing in mind that larger drones with more payload capacity also have several disadvantages, as discussed earlier.

Ideally, the drone should also have additional lights for inspections under the deck and the remote range of the drone must be long enough for inspecting large bridges that are difficult to access. Finally, the drone must be very stable during flight and not react to windy conditions. The drone should vibrate as little as possible during the flight to take high-quality images of the bridge. The vehicle should be equipped with an obstacle-avoidance system to avoid harm to people and property. Furthermore, a GPS is necessary for the autonomous flight of the drone.

Once a reasonable drone and 360° camera are selected, the components must be assembled. In particular, the 360° camera must be firmly mounted on the drone. The drone and the 360° camera are off-the-shelf products, and the component should be easily and reversible build together. The necessary material should be inexpensive and available at any hardware store. Furthermore, the prototype must be very stable in flight and carry the 360° camera without restrictions. It is absolutely important that the camera is fixed very well on the drone for safety reasons. If the camera comes into contact with the propellers during flight, the drone will most likely crash and endanger the operator and other people nearby. In addition, a crash would cause damage to the drone and potentially other objects like parked cars. The drone is also required to fly close enough to the critical parts of the bridge and stabilize in a particular position to take images. Hence, there are high maneuverability requirements for the overall system, and the flight time should not be reduced too much by the additional 360° camera. Due to the budget constraint, the price limit for the 360° camera is set at 500 USD. Table 1 shows the summary of all requirements for the drone, the cameras, and the overall system:

**Table 1.** Overview of System Requirements.

| Component | Requirement | Specification |
|---|---|---|
| Drone | Reasonable flying time | 25–30 min |
| Drone | The light source for dark areas (e.g., below the deck of the bridge) | Additional light source |
| Drone | Reasonable carrying capacity for additional equipment (e.g., 360° camera) | Between 400 and 500 g carrying capacity |
| Drone | Reasonable remote range | Minimum of 500 m remote control range |
| Drone | Cost | Below 2500 USD |
| Drone | Avoiding obstacles | Obstacle avoidance system |
| Drone | Stable flight and positioning (also in windy conditions) | Flight stabilization system |
| Drone camera | High-definition camera with high resolution with low illumination | Resolution of at least 4K |
| Drone camera | No reaction to vibrations of the drone | Camera stabilisation system |
| Drone camera | Camera must be rotatable | Rotatable camera |
| 360° camera | High-definition camera with high resolution with low illumination | Resolution of at least 4K |
| 360° camera | Single picture | Camera must be able to generate a single 360° image |
| 360° camera | Remote control | WiFi remote-controlled camera with real-time display |
| 360° camera | No reaction to vibrations | Camera stabilisation system |
| 360° camera | Cost | Below 500 USD |
| Mounting of 360° camera | Easy to build and reversible | No permanent fixation or use of glue |
| Mounting of 360° camera | 360° camera must be firmly attached | Firm fixation of 360° camera |
| Mounting of 360° camera | Cost | Below 50 USD |
| System | Stable flight with an additional 360° camera | Flight stabilization that works with additional load |
| System | Reasonable flight time with additional equipment | Minimum 20 min |

The requirements for the drone have been discussed and specified. Drones for inspection tasks have been used before by, e.g., [10]. The authors compared and evaluated different off-the-shelf drones with regard to price, flying time, camera resolution, video resolution, payload capacity, light source, and remote range. The drone with the highest rating (5 on a scale between 1 and 5), according to [10], is the SenseFly Albris. However, the drone costs 45,000 USD and is beyond the budget. The drones with a rating of 4 are DJI Matrice 100, DJI Phantom 3 Pro, DJI Phantom 4, and the DJI S900 airframe. As the DJI Matrice 100 and the DJI S900 airframe cost 3000 USD and more, the author will stick to the budget constraint and purchase a drone below the 2500 USD price range. Hence, the DJI Phantom 4 and the DJI Phantom 3 Pro are the only remaining drones below 2500 USD.

Following Seo et al. [10], the DJI Phantom 4 PRO V2.0 is selected as a drone platform for bridge inspection. The selected off-the-shelf drone meets several requirements that are necessary for the bridge inspection task by the drone, as discussed. First, the flying time of the drone is in excess of 25 min and thus allows for a comprehensive inspection. Second, the drone is equipped with a light source to be able to capture high-resolution images with low illumination. Third, the drone has a payload capacity in excess of 450 g to carry additional equipment like the 360° camera. Fourth, the remote range is large enough to access structures that are located over water or are difficult to access by inspectors. Finally, the drone has a camera capable of shooting 4K images and videos at 60 fps, at up to 100 Mbps, and capturing 20-megapixel stills. The DJI Phantom 4 has an obstacle avoidance system that allows the drone to avoid harm to persons, property, and the drone itself. Furthermore, the drone can be operated in manual mode because the GPS signal might be lost under the bridge.

In order to take high-quality images of defects for inspection purposes with a 360° camera, several requirements have been discussed and specified. Different 360° cameras have been recently compared and evaluated by, e.g., [18]. The budget constraint for the off-the-shelf 360° camera was set to 500 USD.

For the bridge inspection, the author considered various 360° cameras. First, the author tested a 4K Actioncam. Generally, the product was able to take decent high-resolution images. However, a major drawback was that the camera produced two 180° images that must be manually put together. During testing, the author found it difficult to fit both images together and decided that the 4K Actioncam was not particularly suited for the purpose of bridge inspection.

As a second choice, the author tested a retail 360° camera that was used in many VR settings. Although the camera produced a 360° image and it was not necessary to manually merge images, the resolution was not good enough for the bridge inspection purpose. Furthermore, the camera was quite sensitive to vibration and produced blurred images during flight.

Finally, the author tested the Ricoh Theta V. The camera was priced at 430 EUR (at the end of September 2020, this is equivalent to ~498 USD and thus below the 500 USD threshold). The camera can be connected to a smartphone via wireless LAN or Bluetooth. Once the camera is connected to the smartphone it can be remote controlled and the current view of the camera is displayed on the smartphone app. The camera has a good battery life of approximately 80 min of video recording or 300 images. The effective pixels are approximately 12 megapixels (x2), and the output is equivalent to 14 megapixels. The camera has a well working image stabilization system. The author, therefore, follows [19–21], and uses the Ricoh Theta V 360° camera for the bridge inspection application.

In order to fix the Ricoh Theta V 360° camera onto the drone, a perforated thin metal sheet was used in combination with cable ties. The camera had a screw thread at the bottom for a tripod screw. With the tripod screw, the camera can be firmly fixed to two metal sheets that were in line with the arms of the drone. The metal sheets can then be firmly fixed to the arms of the drone with the cable ties. The cable ties can be securely tightened between the arms of the drone and the handgrips on the one side and through a hole in the metal sheet on the other side. Figure 3 shows the components for mounting the 360° camera on the drone.

After the fixation of the mounted camera on top of the drone, the system was tested for its stability, and a maiden flight took place. Figure 4 shows the first flight of the DJI Phantom 4 Pro with the Ricoh Theta V 360° camera mounted on top of the drone:

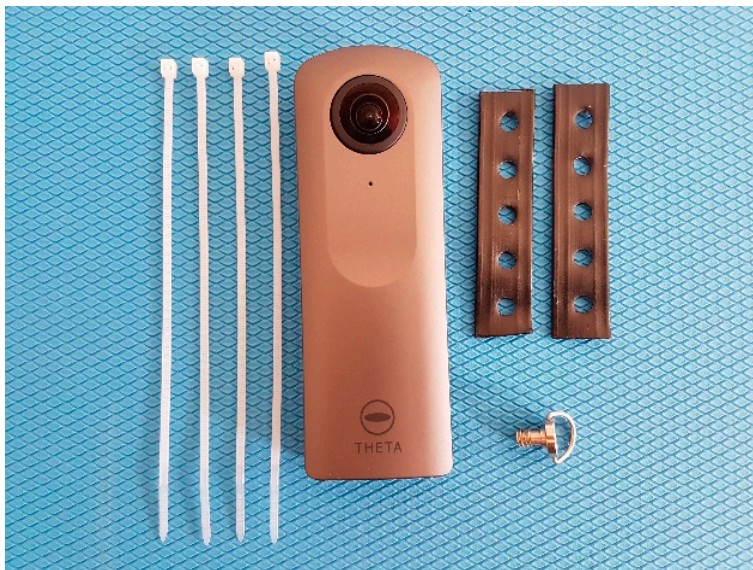

**Figure 3.** Mounting Equipment.

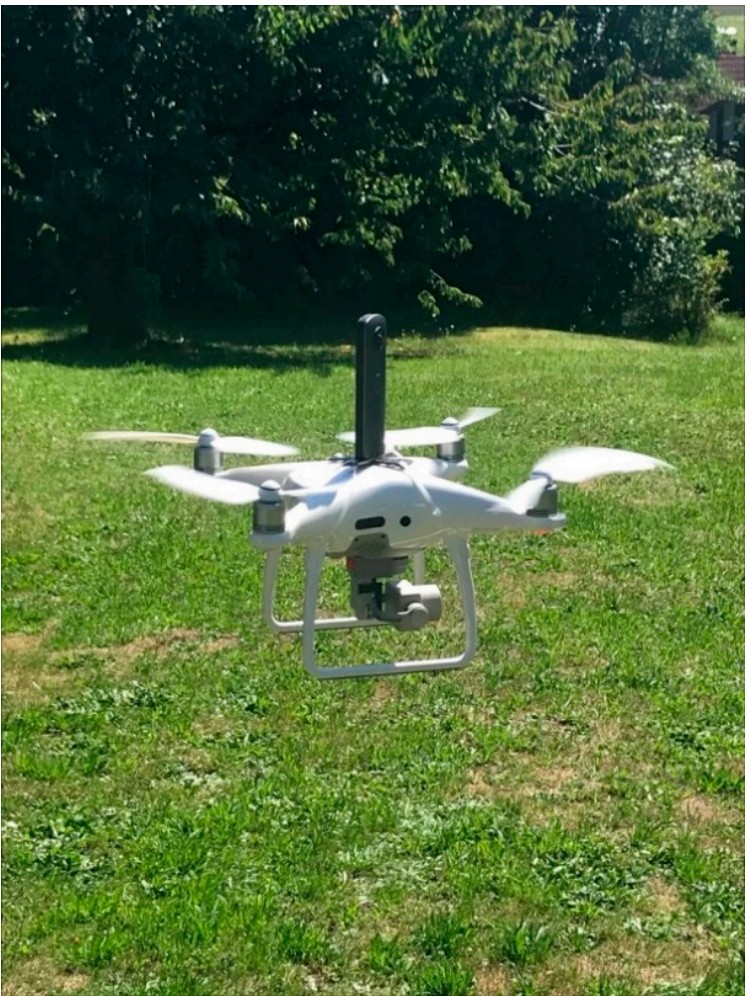

**Figure 4.** Maiden Flight.

Furthermore, the functioning of the Ricoh Theta V 360° camera mounted on the top of the drone was tested. In particular, the quality of the images was verified as drone vibrations might blur the

pictures. The camera was connected to a smartphone and remote-controlled to take images. The 360°
camera worked very well on the flying drone, and the images appeared to be of high quality.

## 4. Bridge Inspection and Image Collection

As discussed earlier, the author followed the bridge inspection methodology developed by [10].
Unfortunately, it was not possible to get any information about the bridges, such as a building plan or
historical inspection reports, and thus the bridge information review was skipped. Next, a site risk
assessment of the bridge was conducted. At both bridge sites, there were many trees nearby that might
cause risk when flying with the drone. Furthermore, at the first bridge, there were train tracks below
the bridge and a bridge deck for cars. Although the train tracks were not in use anymore and there was
no risk of a train accident, there was a risk of stumbling and falling down. Especially when flying the
drone, this might cause risk for not only the operator but also other people and objects if the operator
stumbles and the drone is out of control. Sometimes there were cars crossing the bridge, and the
operator must make sure that the drone does not cause any risk to the cars. However, there were no
power lines or an airport close to the site, and the next buildings were a few hundred meters away.
At the site of the second bridge, there were also cars driving on the road below the bridge, but the train
tracks on the bridge deck were not used anymore. However, there was a small river next to the street
and this at least also causes risk for the drone in any crash scenario. Again, there were no power lines
or an airport close to the site, and the next buildings were a few hundred meters away. The third step
consisted of the drone pre-flight setup. A thorough inspection of the drone and software was executed.
The picture in Figure 5 shows the preparation for drone crack inspection at the first bridge:

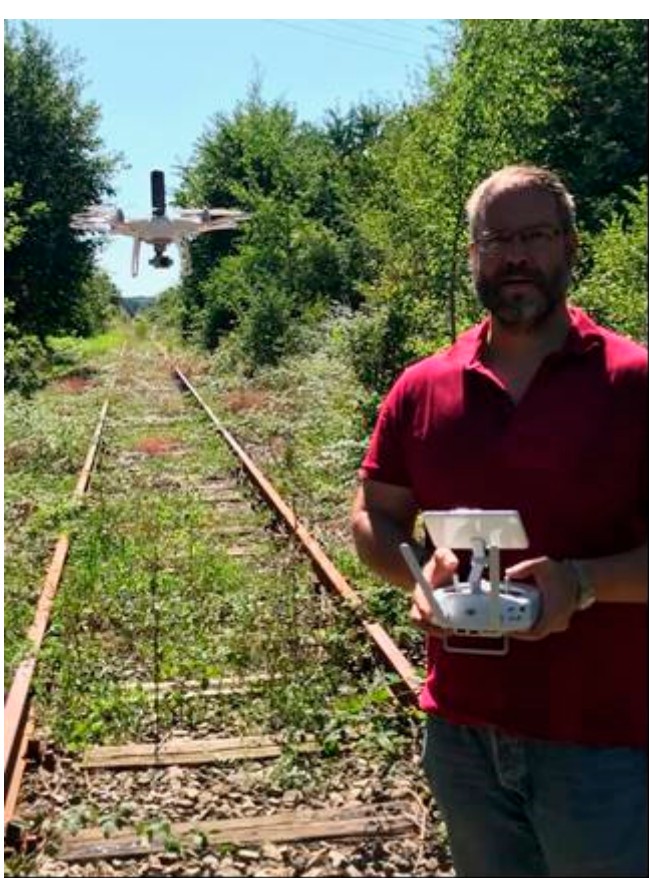

**Figure 5.** Drone Pre-inspection Check.

Although arch bridges appear to be more simplistic in terms of critical parts, the correct symmetry of the arch is very important. For instance, Ye et al. [22] used a method for inspecting arch bridges that rely on mapping deformations and inferring movements of arch bridges.

Figures 6 and 7 illustrate the main components of the two bridges. They consist of the railing, tracks, bearings, superstructure, abutment, and abutment foundation for the first bridge (Figure 6). The second bridge is an arch bridge, and Figure 7 shows the crown and arch barrel of the bridge. Figure 8 shows the bridge in Thambach (second bridge) with various visible defects like corrosion, concrete spalling, waterproofing failure and cracks.

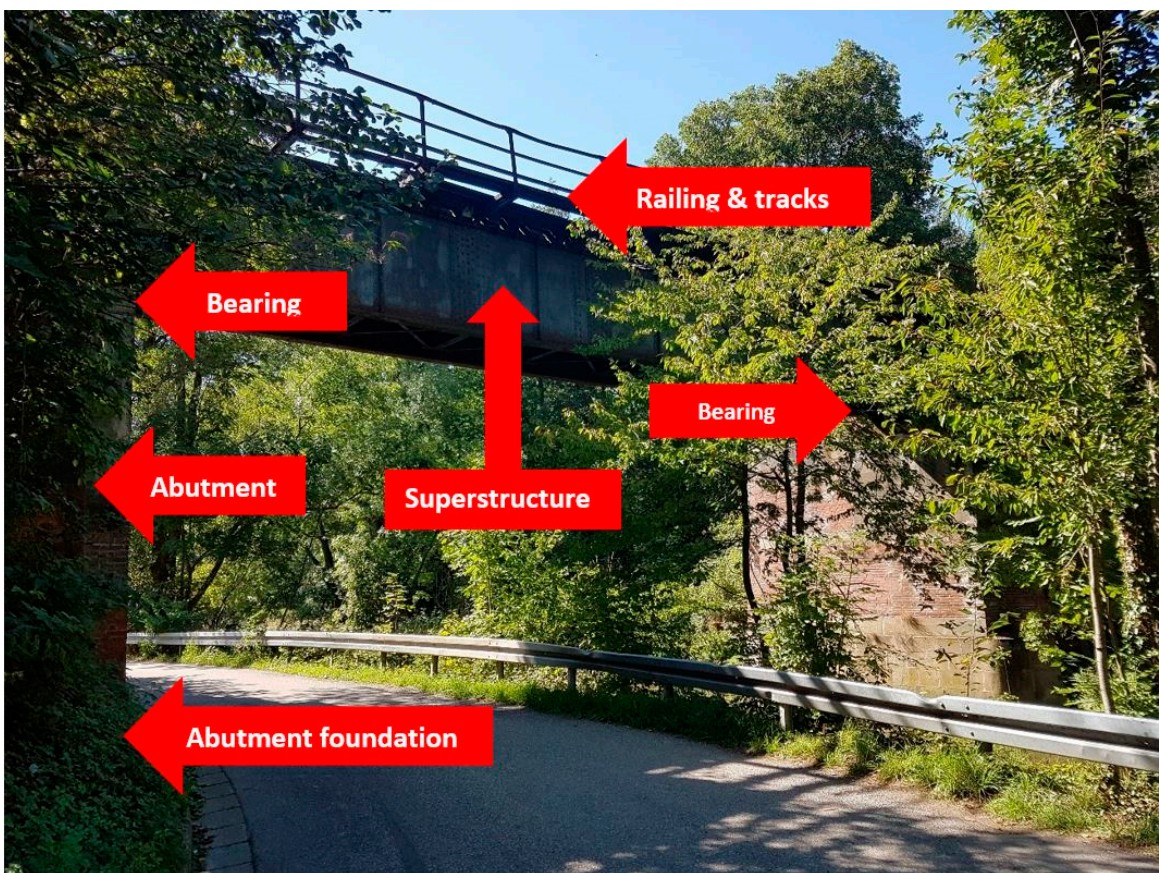

**Figure 6.** Front view of selected bridge in Gangkofen, Germany.

Figure 7 also shows that the ceiling of the bridge cannot be properly observed with the standard version of the drone. To get a 360-degree image of the bridge, the modified drone with a Ricoh Theta V 360° camera on top of the drone was used. For instance, [15] applied a similar approach to a wheeled robot and used a Samsung gear 360-degree camera mounted on the mobile robot for culvert inspection. The Samsung camera has a dual-lens and a single-lens mode that can be used for capturing photos or video. When the Ricoh Theta V 360-degree camera is mounted on top of a drone, a continuous view of the inner walls and the celling of a bridge can be captured while it flies underneath the bridge. Thus, hard to reach areas can be visually inspected. According to [15], the greatest advantage of a 360° camera is that it is not necessary anymore to move, pan, or tilt the camera during inspection.

Figures 9 and 10 show a comparison of images taken from the different component of the bridge by the HD camera underneath the drone and by the 360° camera on top of the drone:

Generally, the HD camera underneath the drone and the 360° camera on top of the drone were both able to take images of the different components of the bridges (see Figures 9 and 10 for details). However, there were two exceptions to this. First, as expected, the camera underneath the drone cannot properly observe the ceiling of the bridge. Second, it was not possible to take images of the

bridge deck with the 360° camera on top of the drone. This might be a major disadvantage when using the proposed inspection method with the 360° camera on top of the drone. The analysis of the bridge bearing also supports the hypothesis that corners can be better observed with the 360° camera on top of the drone compared to the HD camera underneath the drone.

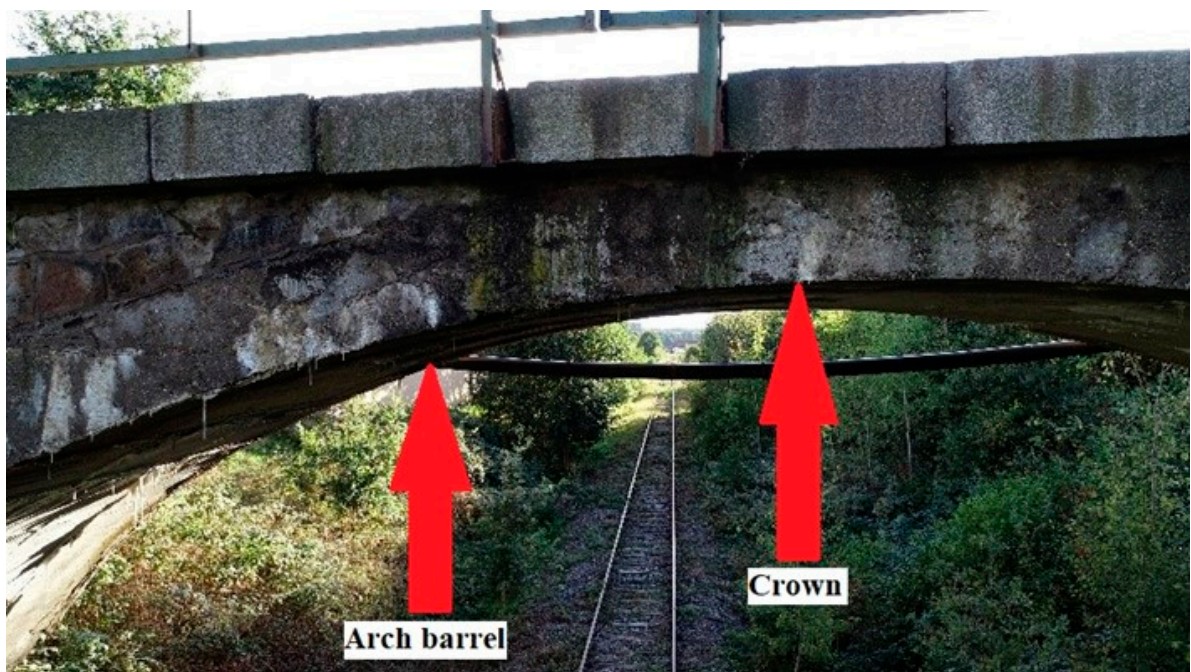

**Figure 7.** Front view of selected bridge in Thambach, Germany.

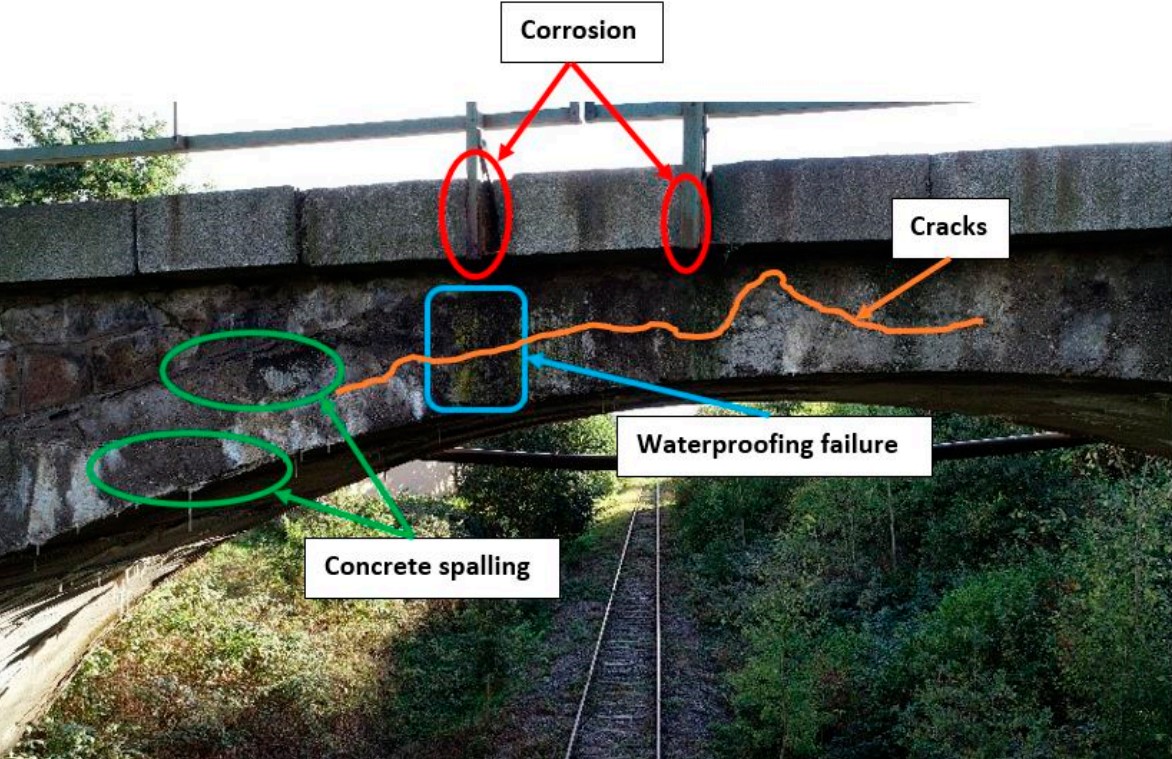

**Figure 8.** Front view with various defects of the selected bridge in Thambach, Germany [23].

| Bridge 1: | | | |
|---|---|---|---|
| Component | HD camera | 360° camera | Comparison |
| Arch and front/back of bridge |  |  | Results appear to be similar. |
| Arch ceiling |  |  | For the ceiling the 360° camera is able to get an image of the whole ceiling. Therefore, the 360° camera is better than the HD camera. |
| Abutment (Pier) and foundation |  |  | Results appear to be similar. |
| Deck |  | It was not possible to get any pictures from the bridge deck with the 360° camera. | Only the HD camera can take images of the deck. |
| Railing and wall |  |  | Results appear to be similar. |

**Figure 9.** Comparison of the HD and 360° camera images for the bridge in Thambach, Germany.

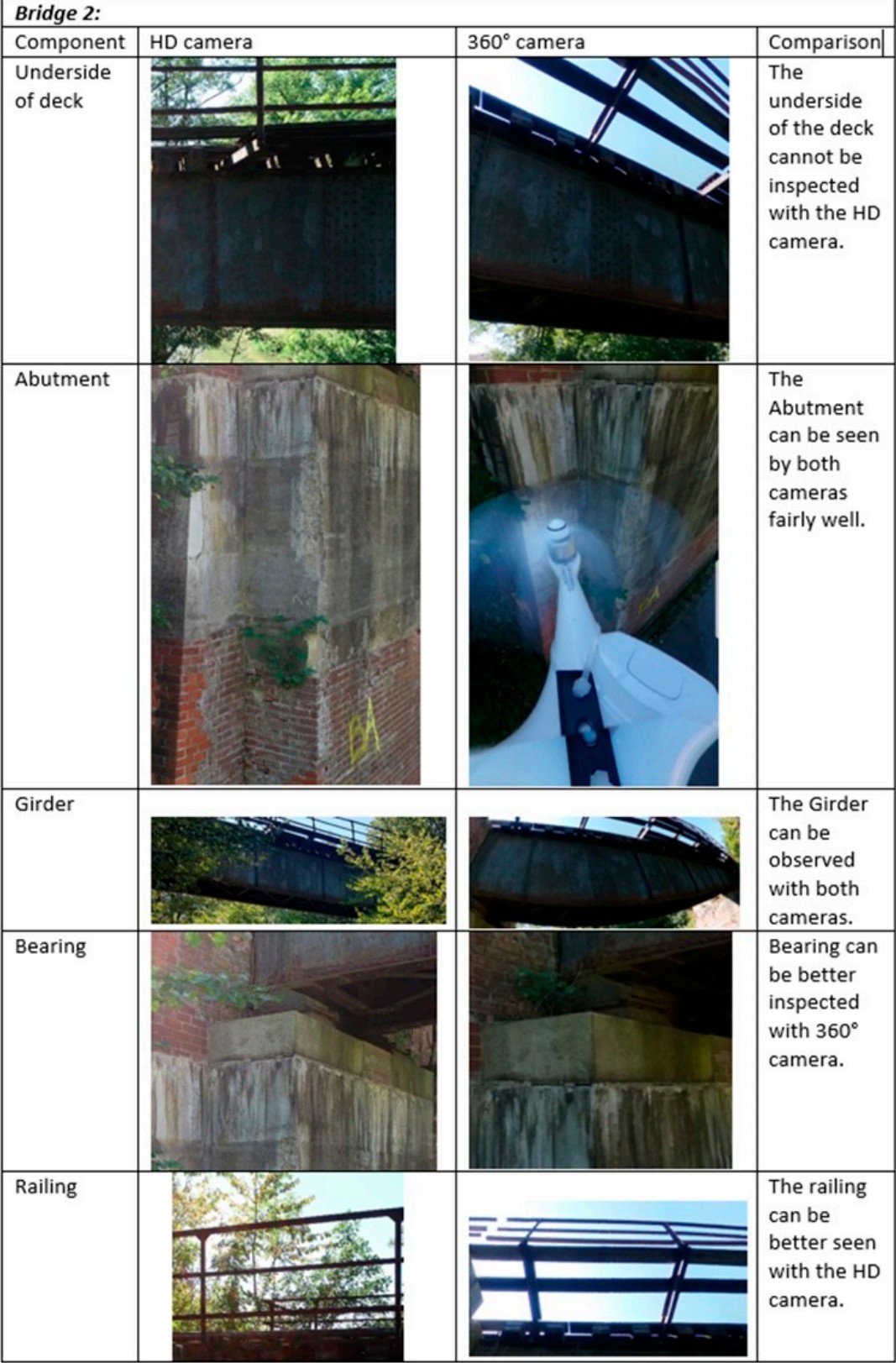

**Figure 10.** Comparison of the HD and 360° camera images for the bridge in Gangkofen, Germany.

The pictures in Figure 11 show different defects at the two different bridge sites. The images clearly show cracks in the concrete, waterproofing failure, concrete spalling, and corrosion of the guardrail.

| Defect | Example | HD vs. 360° camera |
|---|---|---|
| Concrete Spalling | | The concrete spalling can be well observed by both cameras. The image shows concrete spalling at the Abutment of the first bridge. |
| Concrete cracking | | The image shows a concrete crack at the Abutment of the second bridge. This damage was detected by both cameras |
| Efflorescence | | At the first bridge we found efflorescence and discoloration at the arch ceiling. The damage could be observed by both cameras. However, the HD camera underneath the drone cannot take images of the ceiling close to the crown of the arch. |
| Corrosion | | We found corrosion at the railing of both bridges. The corrosion could also be observed by both cameras. |
| Waterproofing failure | | At the first bridge there was evidence of waterproofing failure. This was also detected by both cameras. |

**Figure 11.** Detected defects at the two different bridge sites.

Overall, the manual inspection of the images reveals many different defects in both bridges. Concrete spalling was more evident in bridge number one, whereas concrete cracking was observed in bridge number two. Efflorescence, corrosion, and waterproof failure were also found in both bridges, as can be seen in Figure 11. The author would, therefore, recommend to increase the maintenance of the two bridges and repair the various defects.

Furthermore, the author tried to take pictures of the ceiling of the arch bridge in order to compare the images of the standard HD camera underneath the bridge with the images taken with the 360° camera mounted on top of the drone. The picture on the top of Figure 12 shows that it is almost impossible to take images from the ceiling of the arch bridge with the standard HD camera underneath the drone. However, the two pictures on the bottom of Figure 12 illustrate how well the 360° camera on top of the drone can take images of the ceiling.

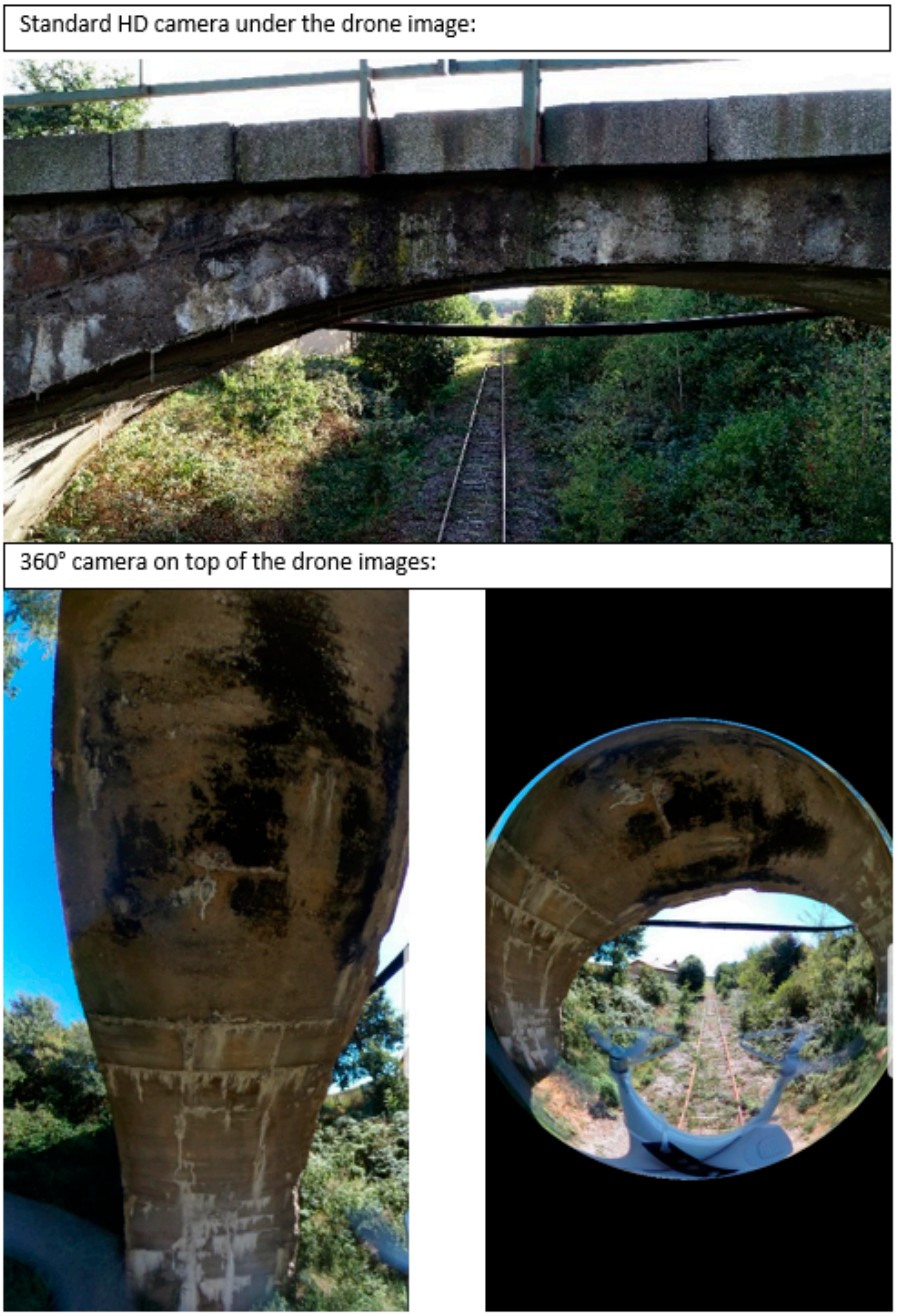

**Figure 12.** Comparison of the HD and 360° camera images of the arch bridge ceiling.

Fortunately, even a closer look at the bridge ceiling does not reveal any major cracks. However, some concrete spalling and waterproofing failure is visible and should be monitored in the future. The author concludes that the 360° camera on top of the drone is much better suited to taking images of the bridge ceiling compared to the standard HD camera underneath the drone.

## 5. Automatic Crack Detection

As an example, the collected pictures are now used for automatic damage identification of critical parts of the bridge by a software tool. The crack detection is implemented with Matlab crack detection code. Although automatic crack detection is not the scope of this article, the example should verify whether crack detection is possible with images from the 360° camera. The code starts with loading the raw image, and then the contrast is stretched. Next, the image is segmented, and standard morphological operations are performed (e.g., thin, clean, and fill). Finally, the Matlab function "imtool" is used for pixel length determination and crack length calculation.

To test the crack detection with the standard HD camera below the drone, an image of a wall with cracks was used. Figure 13 shows on the top the raw picture of the cracks in the wall. The cracks can already be observed very well visually. On the right side in Figure 13, the result of the Matlab crack detection analysis for the HD camera is displayed. The original image is represented by a RGB scale with brightness and color information. Generally, there are various factors that must be taken into account when an RGB image is analyzed directly [24]. For that reason, the RGB image is converted to a grayscale image that only contains brightness information with grayscale values of a pixel between 0 and 255. Larger values represent whiter and brighter pixels, with 255 being pure white and 0 being pure black. The applied algorithm makes use of a thresholding procedure to transform the grayscale image into a binary image only containing black and white pixels. The results of this image segmentation can be found in Figure 13 labeled binary image. The author found a threshold value of 200 for image segmentation works best for crack detection in this example. However, for automatic crack detection, a maximum-entropy algorithm [25] or Otsu's method [26] is recommended. Furthermore, the distance between the camera and the wall was roughly one meter. A shorter distance was not possible due to safety reasons, and a larger distance deteriorates the results. To calculate the length of the crack, Matlab inks the detected defaults in green color, as can be seen in Figure 13 in the row labeled final image. The results show that the Matlab algorithm only detects part of the crack. The calculated long slope of the crack is 0.7499, and the perpendicular slope is 1.3334.

Next, crack detection with the 360° camera is tested by taking a picture of the same wall with cracks at the same distance. Figure 13 on the left shows the raw picture of the cracks in the wall taken by the 360° camera on the top. The cracks can be observed visually. Again, the lower images in Figure 13 on the left show the result of the Matlab crack detection analysis.

In comparison to the standard HD image, the automatic code detection in the 360° camera image appears to improve. The calculated long slope of the crack is 0.8157, and the perpendicular slope is 1.2260. The cracks in the wall are almost fully observed by the algorithm. More examples of crack detection in the 360° camera images are shown in Figure 14. Although the algorithm generally finds the cracks in the surface, most cracks are not fully observed by the algorithm. Further research is necessary to improve the crack detection. However, the examples support the use of a 360° camera for crack detection as the results are similar to the HD camera under the drone.

As the results are only based on a few examples, the findings cannot be generalized. To generalize this conclusion, future research must use a more rigorous analysis that compares many images of different cracks.

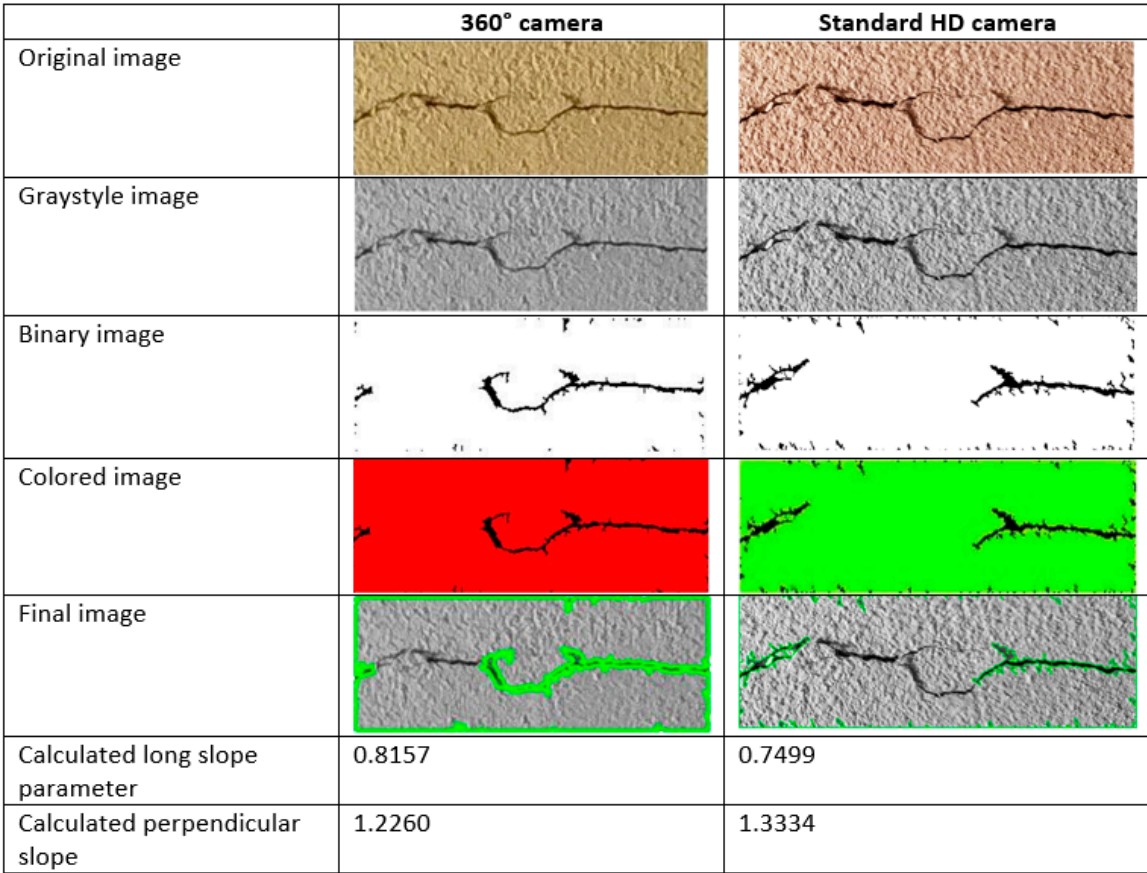

| | 360° camera | Standard HD camera |
|---|---|---|
| Original image | | |
| Graystyle image | | |
| Binary image | | |
| Colored image | | |
| Final image | | |
| Calculated long slope parameter | 0.8157 | 0.7499 |
| Calculated perpendicular slope | 1.2260 | 1.3334 |

**Figure 13.** Crack detection with a 360° camera and standard HD camera.

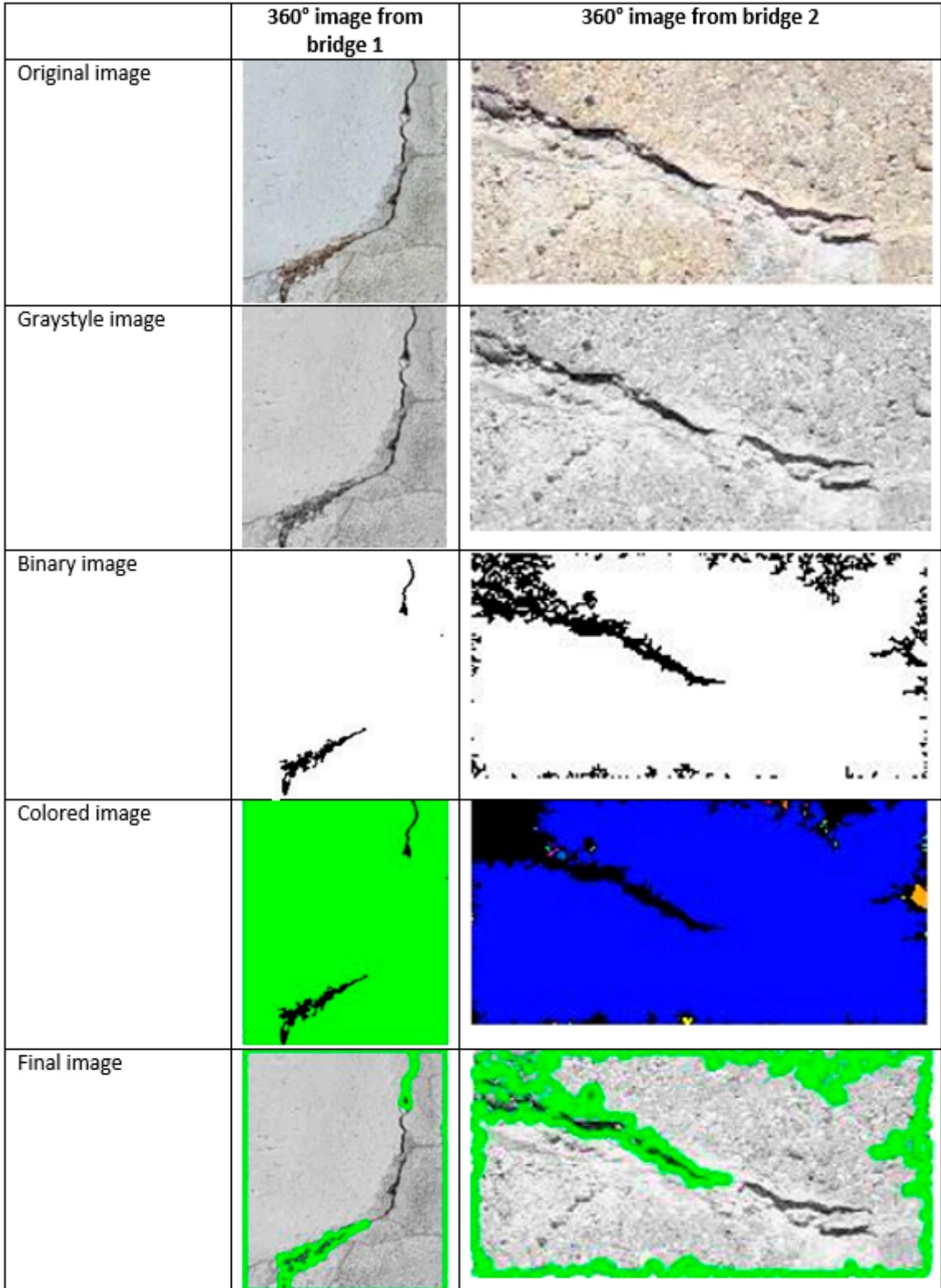

**Figure 14.** Crack detection with the 360° camera at bridge one and two.

## 6. Discussion

The author successfully managed to use an off-the-shelf drone with an off-the-shelf 360° camera on top of the vehicle for bridge inspection. This generally supports earlier studies by inter alia [9,10,12]

that all used drones for bridge inspection. Seo et al. [10] used an off-the-shelf DJI Phatom 4 without modifications to inspect a timber girder bridge. In contrast, Sanchez-Cuevas et al. [12] used a self-made drone with an ultrasonic sensor that required physical contact with the surface of the bridge, whereas Lee et al. [9] applied a Leica Aibot drone system that costs 50,000 USD. Therefore, many researchers have either used self-made special-purpose drones or very expensive industrial drones. The approach in this paper bears the advantage of being inexpensive and easily applied. The operator does not have to assemble the drone system, can purchase the equipment at an internet store, does not need substantial technical training, and the lead time is very short. In contrast to studies that also used an off-the-shelf drone, the author further equipped the drone with a 360° camera on top of the vehicle and applied it to, e.g., an arch bridge.

The bridge inspections were limited to visual inspection of bridges with a HD camera underneath the drone and a 360° camera on top of the drone. Visual bridge inspection with a drone is the most common use of drone inspections, and the author supports earlier work by other researchers. However, the use of 360° cameras is very limited, and the author is one of the first to apply it to crack detection in bridges. In that respect, the article adds to the literature by showing the successful use of an off-the-shelf 360° camera for crack detection purposes.

Critical parts of bridges for inspection include corners and the ceiling of the bridge deck. As earlier studies mainly used cameras underneath the drone, it is not possible or at least very difficult to take images of a ceiling or corners with this approach. The author proposes a natural solution to this problem by using a 360° camera on top of the drone that is very well suited to taking images of areas above or diagonally above the drone. In particular, the author shows the drone based visual inspection of an arch bridge. Especially arch bridge inspection with drones is very limited in the literature and there are special considerations when inspecting arch bridges. Overall, it seems reasonable to combine the use of a 360° camera on top of the drone with the camera underneath the drone for proper bridge inspection.

Visual crack detection with digital images has been researched extensively, and much progress in automatic crack detection has been made recently. One limitation of the research is the poor automatic or algorithmic crack detection of the presented approach. It was difficult and sometimes not possible to correctly find and calculate the cracks. This is partially due to the need to set threshold values for crack detection manually. However, there exist powerful algorithms today that are much better in correctly detecting cracks.

Traditionally, cracks have been inspected manually by, e.g., visually observing the surfaces and structures of, e.g., a bridge. However, the outcome of the manual inspection completely depends on the knowledge and experience of the specialist inspecting the bridge. Hence, automatic or algorithmic images based crack detection has been proposed to increase the objectivity and accuracy of crack detection. Engineering structures with concrete surfaces or beams are often affected by fatigue stress and cyclic loading that leads to cracks that usually initiate at the microscopic level on the surface of the structure [27]. The objective of automatic image crack detection is to identify the type, number, width, depth, and length of cracks in the surface. This helps to determine early degradation as well as the carrying capacity of the concrete.

For instance, [28] developed an automated crack detection algorithm for crack detection on concrete bridges. They point to the importance of crack detection on bridge decks for maintaining the structural health and reliability of the concrete. They applied a machine learning classification algorithm and thus eliminated the need for manually tuning threshold parameters.

The authors concluded that the algorithm showed 90% accuracy on thousands of test cracks. The algorithm was tested and trained with geographically separate datasets and evaluated on real-world data.

Yamaguchi et al. [29] stated that many automated crack detection systems fail because the image of the concrete surface contains various types of noise due to different causes such as concrete blebs, stain, insufficient contrast, and shading. Earlier studies, for instance, applied a Canny filter and

the wavelet transform for crack detection (see Abdel-Qader et al. [30] and Hutchinson et al. [31]). Yamaguchi et al. [29] followed a similar approach by using a receiver operating characteristics analysis. Further, they improved the crack detection system by noise reduction based on the percolation model. Overall, they reported high accuracy in detecting cracks in concrete surfaces. For a critical review on automatic crack detection, see inter alia Kaur et al. [32] and Patil et al. [33].

Finally, the author was able to take sharp images under the bridge. However, with a large bridge, the light conditions might be very poor, and an additional light source might be needed for inspection. Poor or uneven illumination might be problematic for crack detection. For a discussion regarding crack detection under poor illumination, see inter alia Su et al. [34]. The authors propose a stationary wavelet transformation to remove the influence of uneven illumination. Furthermore, for segmentation, they improved pulse coupled neural network algorithm based on a Markov network's gray threshold that directly completes the segmentation without the need to set parameters manually.

For larger bridges, a major issue is how to navigate the drone under complex bridge structures and how to locate defects from the perspective of the bridge coordinate system. For instance, Guerrero et al. [35] applied a Zermelo-Traveling Salesman Problem method to compute the optimal route to inspect a bridge structure. In this approach, the inspection coordinates for the interest points were obtained automatically by means of a meshing algorithm. The Zermelo-TSP method was used to compute the time-optimal route to inspect the bridge at the relevant points. Similarly, Hallermann et al. [36] implemented a GPS-preplanned automatic displacement analysis based on photogrammetric computer vision methods. They suggested that this inspection method, with the possibility of geo-referencing of images, offered an efficient and accurate way for inspection of large scale structures like bridges. However, in practice, optimal flight path and automated defect location in bridges were still immature, and further research is needed for fully automated bridge inspection.

Although the author found stable flight behavior of the modified drone, this might be different in more windy conditions often found around large bridges. The reason for placing a camera under the drone's frame is to keep the center of gravity under the drone and increase stability. Positioning a camera above the drone ultimately decreases stability, and this might be hazardous in windy environments. Future research must, therefore, test flight stability in turbulent environments.

## 7. Conclusions

The overall aim was to assess the application of an off-the-shelf drone with a 360° camera mounted on the top of it for bridge inspection purposes. Given a budget constraint, components that fulfill the requirements were selected. The final system consists of a DJI Phantom 4 Pro drone and a Ricoh Theta V 360° camera. The author managed to find two relevant bridges to evaluate the use of the 360° camera on top of a drone for bridge inspection. The total cost of the drone inspection systems is below 3000 USD and, therefore, also affordable for smaller municipals or in poor countries.

The objective was to prove that an off-the-shelf drone can be easily equipped with an off-the-shelf 360° camera in order to be used for bridge inspection. A major concern was that vibrations emitted by the drone might blur the camera images and make it impossible to detect cracks. However, contrary to expectations, vibrations and blurred images were not a problem.

The author was able to show that the cracks that were identified with a conventional HD camera could also be identified with the 360° camera. In the beginning, the author was concerned that the use of a 360° camera might make it more difficult to identify some cracks because a 360° camera uses a special lens that takes round pictures and bends straight lines. However, the camera automatically transfers the round pictures to "normal" shapes on a 360° image. Overall, this process did not deteriorate the quality of the image and had no negative impact on crack detection. However, to generalize this conclusion, future research analyzing many images is necessary.

The author was even able to show that the 360° camera on top of the drone can detect defects that were not detected by the conventional approach. The reason is that especially the ceiling cannot be observed with a camera below the drone, whereas a 360° camera on top of the drone is able to observe

the ceiling and detect defects. Besides the problem of potentially bended raw images, the author also expected poor light conditions might have negative effects when inspecting a ceiling. However, in the two case studies, the quality of the images was good, given the poor light conditions. Furthermore, the 360° camera on top of the drone was much better suited to take images of corners (e.g., as evident when inspecting the bridge bearing of the bridge in Gangkofen) and round arches compared to the conventional approach with a camera underneath the drone.

This research adds to the literature as the author is one of the first to attach an off-the-shelf 360° camera to an off-the-shelf drone for bridge inspection. Furthermore, the research shows the advantage of a 360° camera on top of a drone compared to a standard HD camera underneath the drone. In particular, critical areas for crack inspection include arches, corners and the ceiling of a bridge, and a 360° camera above a drone is better suited to access those areas compared to an HD camera underneath a drone.

**Funding:** This work was financially supported through the Open Access Publication Fund of the Munich University of Applied Sciences.

**Acknowledgments:** This article is partially based on a master thesis written by the author at Brunel University, London. I would, therefore, like to thank my supervisors Mujib Rahman and Alireza Mousavi, for the vulnerable and helpful comments and support. Furthermore, I would like to thank the reviewers for their helpful and constructive reviews that helped to substantially improve the paper.

**Conflicts of Interest:** The author declares no conflict of interest.

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
