# Peer review of "Bridge Inspection with an Off-the-Shelf 360° Camera Drone"

_drones, doi:10.3390/drones4040067_

Round 1

Reviewer 1 Report

This work deals the visual inspection monitoring with drones to evaluate damages in civil infrastructure. Specifically, several scenarios such as bridges are considered to evaluate the changes proposed to the traditional inspection with drones.
I suggest to the author review the following comment for improving the work:
-Although in the paper there is only one author, all the work is written in the plural (We).
-Sections funding, acknowledgments, conflict of interest are exactly as in the original format from mdpi. In this way, the work seems like a draft previous to the final submission.
-It is not clear the main objective of the paper. In data collection and prototype design, it is remarked the design of the drone but actually, the design is oriented to choice the camera and the use of a commercial platform. Is it correct? -The major contribution is to define the position and the kind of camera to use in the prototype. I suggest emphasizing these aspects and others that you consider that can highlight the proposal design.
-Multiple images were acquired with HD and 360° camera showing the advantages of the changes proposed for the visual inspection, however, the results are poor in the use of automatic crack detection. I suggest to include more results and emphasize in aspects such measurements to evaluate false detection in terms of acquired images

Author Response

Dear Reviewer 1,

Thank you very much for your helpful comments and suggestions. I have implement all of your recommendations and think this very much improved the article. I made the following changes with regard to your suggestions:

Q1: Although in the paper there is only one author, all the work is written in the plural (We)

A1: You are right – I used we and wrote the article alone. I know this is a German thing, but we do not use the word “I” in academic articles. Therefore I changed the word “We” now to “the author” in the article. I hope this is fine as it is singular now.

Q2: Sections funding, acknowledgments, conflict of interest are exactly as in the original format from mdpi. In this way, the work seems like a draft previous to the final submission.

A2: This was a mistake and I changed it:

Funding: This work was financially supported through the Open Access Publication fund of the Munich University of Applied Sciences.

Acknowledgments: This article is partially based on a master thesis written by the author at Brunel University, London. I would therefore like to thank my supervisors Prof. Dr. Mujib Rahman and Dr Alireza Mousavi for the vulnerable and helpful comments and support. Furthermore, I would like to thank the reviewers for their helpful and constructive reviews that helped to substantially improve the paper.

Conflicts of Interest: The author declares no conflict of interest.

Q3: It is not clear the main objective of the paper. In data collection and prototype design, it is remarked the design of the drone but actually, the design is oriented to choice the camera and the use of a commercial platform. Is it correct? -The major contribution is to define the position and the kind of camera to use in the prototype. I suggest emphasizing these aspects and others that you consider that can highlight the proposal design.

A3: Thank you very much for pointing this out. To clarify this, I added a section with the objective beginning in line 97:

The main objective of the paper is to define the position and the kind of camera to use in the prototype for bridge inspection. Furthermore, the article investigates whether a 360° camera on top of a drone can produce images for automatic crack detection comparable to images from a standard HD camera underneath the drone.

Q4: Multiple images were acquired with HD and 360° camera showing the advantages of the changes proposed for the visual inspection, however, the results are poor in the use of automatic crack detection. I suggest to include more results and emphasize in aspects such measurements to evaluate false detection in terms of acquired images

A4: I absolutely agree and added another image as well as included more detail regarding the crack detection process. However, as you pointed out in Q3, the objective of the paper is not focused on crack detection and this section should “only” show that it is possible in principle with a 360° camera. The detailed crack detection with this camera is the subject of another paper and is generally very complex. Therefore, this is out of the scope of the current paper. Nonetheless, I added substantial information in the relevant section (including more cracks in Figure 13) and hope you will appreciate the current version on crack detection:

As an example, the collected pictures are now used for automatic damage identification of critical parts of the bridge by a software tool. The crack detection is implemented with Matlab crack detection code. Although automatic crack detection is not the scope of this article, the example should verify whether crack detection is possible with images from the 360° camera. The code starts with loading the raw image and then the contrast is stretched. Next, the image is segmented and standard morphological operations are performed (e.g. thin, clean and fill). Finally, the Matlab function “imtool” is used for pixel length determination and crack length calculation.

To test the crack detection with the standard HD camera below the drone, an image of a wall with cracks was used. Figure 13 shows on the top the raw picture of the cracks in the wall. The cracks can be already observed very well visually. On the right side in Figure 13, the result of the Matlab crack detection analysis for the HD camera is displayed. The original image is represented by a RGB scale  with brightness and colour information. Generally, there are various factors that must be taken into account when a RGB image is analyzed directly [24]. For that reason, the RGB image is converted to a grayscale image that only contains brightness information with grayscale values of a pixel between 0 and 255. Larger values represent whiter and brighter pixels with 255 being pure white and 0 being pure black. The applied algorithm makes use of a thresholding procedure to transform the grayscale image into a binary image only containing black and white pixels. The results of this image segmentation can be found in Figure 13 labelled binary image. The author found a threshold value of 200 for image segmentation works best for crack detection in this example. However, for automatic crack detection a maximum-entropy algorithm [25] or Otsu’s method [26] is recommended. Furthermore, the distance between the camera and the wall was roughly one meter. A shorter distance was not possible due to safety reasons and a larger distance deteriorates the results. To calculate the length of the crack, Matlab inks the detected defaults in green colour as can be seen in Figure 13 in the row labelled final image. The results show that the Matlab algorithm only detects part of the crack. The calculated long slope of the crack is 0.7499 and the perpendicular slope is 1.3334.

Next, crack detection with the 360° camera is tested by taking a picture of the same wall with cracks at the same distance. Figure 13 on the left shows the raw picture of the cracks in the wall taken by the 360° camera on the top. The cracks can be observed visually. Again, the lower images in Figure 13 on the left show the result of the Matlab crack detection analysis.

Figure 13. Crack detection with 360° camera and standard HD camera (see pdf for the figure)

In comparison to the standard HD image, the automatic code detection in the 360° camera image appears to improve. The calculated long slope of the crack is 0.8157 and the perpendicular slope is 1.2260. The cracks in the wall are almost fully observed by the algorithm. More examples of crack detection in the 360° camera images are shown in Figure 14. Although the algorithm generally finds the cracks in the surface, most cracks are not fully observed by the algorithm. Further research is necessary to improve the crack detection. However, the examples support the use of a 360° camera for crack detection as the results are similar to the HD camera under the drone.

Figure 14. Crack detection with 360° camera at bridge one and two (see pdf for the figure)

As the results are only base on few examples, the findings cannot be generalized. To generalize this conclusion, future research must use a more rigorous analysis that compares many images of different cracks.

Reviewer 2 Report

The paper studies the use of off-the-shelf drones to inspect bridges. This reviewer has been inspecting bridges for several years during his professional career, mainly on railroad lines, and fully agrees with the author that it is necessary to reduce the cost of inspections, given the large number of bridges to be inspected, many of them with areas of very difficult access, such as the arch bridges mentioned in the paper.
This paper is, in my opinion, very interesting, for its practical character. The fact of referring to trademarks or budgets (something rare in technical papers) gives the paper a very high utility for researchers in similar problems.
In my opinion, the paper is fully within the scope of the journal and should be published as it is, provided the following comments are properly addressed
Although this reviewer's mother tongue is not English, some expressions are unfamiliar. In addition, there are some typos (such as "bride" instead of "bridges", in line 196).The language and style should be reviewed.
The Sections "Funding", "Acknowledgments" and "Conflict of interests" must be properly fulfilled (Lines 425-438)

Author Response

Dear Reviewer 2,

Thank you very much for your supportive review. I am very happy to hear that you very much appreciate the work from a practical point of view. This was the main purpose of the paper. Furthermore, I implemented all your suggested changes.

Q1: Although this reviewer's mother tongue is not English, some expressions are unfamiliar. In addition, there are some typos (such as "bride" instead of "bridges", in line 196).The language and style should be reviewed.

A1: Thank you very much for pointing this out. I have made the changes and asked a native and English professor at my University to proof read the paper. The necessary changes have been made.

Q2: The Sections "Funding", "Acknowledgments" and "Conflict of interests" must be properly fulfilled (Lines 425-438).

A2: This was a mistake and I changed it as follows:

Funding: This work was financially supported through the Open Access Publication fund of the Munich University of Applied Sciences.

Acknowledgments: This article is partially based on a master thesis written by the author at Brunel University, London. I would therefore like to thank my supervisors Prof. Dr. Mujib Rahman and Dr Alireza Mousavi for the vulnerable and helpful comments and support. Furthermore, I would like to thank the reviewers for their helpful and constructive reviews that helped to substantially improve the paper.

Conflicts of Interest: The author declares no conflict of interest.

Reviewer 3 Report

The author reports an interesting application of drones for bridge inspection. The highlight is using a 360 degree camera on the top of the drone. Basically, drones have been intensively investigated for various kinds of bridges in recent years. But it still remains great challenging in practice. For example, how to navigate the drone under complex bridge structures, how to automatically identify defects from images, how to handle illumination changes, how to locate defects from a perspective of bridge coordinate system. Unfortunately, these aspects are not involved in this word. It could be more scientifically beneficial if the author goes deeper into these issues. Therefore, the referee thinks this paper cannot be accepted in its current form. - Please annotate which picture is captured by which camera in Figure 12. - What’s the difference of HD camera underneath the drone and the 360 camera on the top of the drone? It’s obvious that the HD camera is not convenient to take picture of the ceiling of arch bridge. Any other difference? For instance, image distortion? - Automatic crack detection algorithm should be elaborated in Section 5. A solid comparison of defect detection results between different cameras and different algorithms is needed. - What’s the distance between lens and bridges? It could impact the size of objects in the pictures. - What’s the challenge in the implementation of bridge inspection? And what’s the solution?

Author Response

Dear Reviewer 3,

Thank you very for your critical and constructive review. I tried to implement all your comments and think this helped to improve the paper substantially.

Q1: For example, how to navigate the drone under complex bridge structures, how to automatically identify defects from images, how to handle illumination changes, how to locate defects from a perspective of bridge coordinate system. Unfortunately, these aspects are not involved in this word. It could be more scientifically beneficial if the author goes deeper into these issues.

A1: I totally agree that the mentioned issues are very important and still discussed in contemporary research articles. However, these issues are not the main objective of the paper and can only be covered extensively in another article. However, I have added a section on this issues and included the experience and state of the current literature on this issues. I hope this is helpful and clears your concerns:

Traditionally, cracks have been inspected manually by e.g. visually observing the surfaces and structures of e.g. a bridge. However, the outcome of the manual inspection completely depends on the knowledge and experience of the specialist inspecting the bridge. Hence, automatic or algorithmic images based crack detection has been proposed to increase the objectivity and accuracy of crack detection. Engineering structures with concrete surfaces or beams are often affected by fatigue stress and cyclic loading that leads to cracks that usually initiate at the microscopic level on the surface of the structure [27]. The objective of automatic image crack detection is to identify the type, number, width, depth and length of cracks in the surface. This helps to determine early degradation as well as the carrying capacity of the concrete.

For instance, [28] developed an automated crack detection algorithm for crack detection on concrete bridges. They point to the importance of crack detection on bridge decks for maintaining the structural health and reliability of the concrete. They apply a machine learning classification algorithm and thus eliminate the need for manually tuning threshold parameters.

The authors conclude that the algorithm shows 90% accuracy on thousands of test cracks. The algorithm was tested and trained with geographically separate datasets and evaluated on real world data.

Yamaguchi, et al. [29] state that many automated crack detection systems fail because the image of the concrete surface contains various types of noise due to different causes such as concrete blebs, stain, insufficient contrast, and shading. Earlier studies for instance applied a Canny filter and the wavelet transform for crack detection (see Abdel-Qader et al. [30] and Hutchinson et al. [31]). Yamaguchi et al. [29] follow a similar approach by using a receiver operating characteristics analysis. Further, they improve the crack detection system by noise reduction based on the percolation model. Overall, they report high accuracy in detecting cracks in concrete surfaces. For a critical review on automatic crack detection see inter alia Kaur et al. [32] and Patil et al. [33].

Finally, the author was able to take sharp images under the bridge. However, with a large bridge the light conditions might be very poor and an additional light source might be needed for inspection. Poor or uneven illumination might be problematic for crack detection. For a discussion regarding crack detection under poor illumination see inter alia Su et al. [34]. The authors propose a stationary wavelet transformation to remove the influence of uneven illumination. Furthermore, for segmentation they us an improved pulse coupled neural network algorithm based on the gray threshold of a Markov network that directly completes the segmentation without the need to manually set parameters.

For larger bridges, a major issue is how to navigate the drone under a complex bridge structures and how to locate defects from a perspective of bridge coordinate system. For instance, Guerrero et al. [35] apply a Zermelo-Traveling Salesman Problem method to compute the optimal route to inspect a bridge structure. In this approach, the inspection coordinates for the interest points are obtained automatically by means of a meshing algorithm. The Zermelo-TSP method is used to compute the time-optimal route to inspect the bridge at the relevant points. Similarly, Hallermann et al. [36] implement a GPS-preplanned automatic displacement analysis based on photogrammetric computer vision methods. They suggest that this inspection method with the possibility of geo-referencing of images offers an efficient and accurate way for inspection of large scale structures like bridges. However, in practice optimal flight path and automated defect location in bridges is still immature and further research is needed for fully automated bridge inspection.

Q2: Please annotate which picture is captured by which camera in Figure 12.

A2: I have now included the necessary annotation in Figure 12.

Q3: What’s the difference of HD camera underneath the drone and the 360 camera on the top of the drone? It’s obvious that the HD camera is not convenient to take picture of the ceiling of arch bridge. Any other difference? For instance, image distortion?

A3: The 360° camera was surprisingly good and I could not detect any shortcomings in comparison to the HD camera under the drone. As I mention in the article, I expected that there might be issues with vibrations and as you suggest distortions. For the HD camera under the drone there are shock absorbers and the 360° camera on top of the drone was firmly attached to the drone. The resolution of the two cameras is also comparable and the results were surprisingly good. However, as discussed in the article, I had to test various cameras to find a good solution. As explained in the text, other 360° cameras had problems with distortion, vibrations, poor resolution and so on.

Q4: Automatic crack detection algorithm should be elaborated in Section 5. A solid comparison of defect detection results between different cameras and different algorithms is needed. - What’s the distance between lens and bridges? It could impact the size of objects in the pictures. - What’s the challenge in the implementation of bridge inspection? And what’s the solution?

A4: I completely revised the section and implemented your suggestions. However, this paper is not on automatic crack detection and an extensive analysis is out of scope of this paper. I once again state that automatic crack detection is not the scope of the paper and the example should only show, that crack detection might be possible with a 360° camera as the results are comparable to the HD camera under the drone. However, I added references to automatic crack detection algorithms and literature. The distance between the lens and bridge was set at one meter. A closer distance was not possible due to safety reasons and a larger distance deteriorates the quality of the images (and makes crack detection more difficult). Regarding challenges and solutions I added text and literature under Q1 (see above). Below is the extended section 5.

As an example, the collected pictures are now used for automatic damage identification of critical parts of the bridge by a software tool. The crack detection is implemented with Matlab crack detection code. Although automatic crack detection is not the scope of this article, the example should verify whether crack detection is possible with images from the 360° camera. The code starts with loading the raw image and then the contrast is stretched. Next, the image is segmented and standard morphological operations are performed (e.g. thin, clean and fill). Finally, the Matlab function “imtool” is used for pixel length determination and crack length calculation.

To test the crack detection with the standard HD camera below the drone, an image of a wall with cracks was used. Figure 13 shows on the top the raw picture of the cracks in the wall. The cracks can be already observed very well visually. On the right side in Figure 13, the result of the Matlab crack detection analysis for the HD camera is displayed. The original image is represented by a RGB scale  with brightness and colour information. Generally, there are various factors that must be taken into account when a RGB image is analyzed directly [24]. For that reason, the RGB image is converted to a grayscale image that only contains brightness information with grayscale values of a pixel between 0 and 255. Larger values represent whiter and brighter pixels with 255 being pure white and 0 being pure black. The applied algorithm makes use of a thresholding procedure to transform the grayscale image into a binary image only containing black and white pixels. The results of this image segmentation can be found in Figure 13 labelled binary image. The author found a threshold value of 200 for image segmentation works best for crack detection in this example. However, for automatic crack detection a maximum-entropy algorithm [25] or Otsu’s method [26] is recommended. Furthermore, the distance between the camera and the wall was roughly one meter. A shorter distance was not possible due to safety reasons and a larger distance deteriorates the results. To calculate the length of the crack, Matlab inks the detected defaults in green colour as can be seen in Figure 13 in the row labelled final image. The results show that the Matlab algorithm only detects part of the crack. The calculated long slope of the crack is 0.7499 and the perpendicular slope is 1.3334.

Next, crack detection with the 360° camera is tested by taking a picture of the same wall with cracks at the same distance. Figure 13 on the left shows the raw picture of the cracks in the wall taken by the 360° camera on the top. The cracks can be observed visually. Again, the lower images in Figure 13 on the left show the result of the Matlab crack detection analysis.

Figure 13. Crack detection with 360° camera and standard HD camera (see pdf for the figure)

In comparison to the standard HD image, the automatic code detection in the 360° camera image appears to improve. The calculated long slope of the crack is 0.8157 and the perpendicular slope is 1.2260. The cracks in the wall are almost fully observed by the algorithm. More examples of crack detection in the 360° camera images are shown in Figure 14. Although the algorithm generally finds the cracks in the surface, most cracks are not fully observed by the algorithm. Further research is necessary to improve the crack detection. However, the examples support the use of a 360° camera for crack detection as the results are similar to the HD camera under the drone.

Figure 14. Crack detection with 360° camera at bridge one and two (see pdf for the figure)

As the results are only base on few examples, the findings cannot be generalized. To generalize this conclusion, future research must use a more rigorous analysis that compares many images of different cracks.

Round 2

Reviewer 1 Report

All my questions and suggestions were addressed in this new version of the paper.

Reviewer 3 Report

The authors have addressed some of my concerns. It can be accepted in its current form.